# Rwin-FPN++: Rwin Transformer with Feature Pyramid Network for Dense Scene Text Spotting

**Chengbin Zeng** [1,2,*] **, Yi Liu** [1] **and Chunli Song** [1]

1 School of Big Data, Guizhou Institute of Technology, No. 1, Caiguan Road, Yunyan District, Guiyang 550003, China
2 Key Laboratory of Electric Power Big Data of Guizhou Province, Guiyang 550003, China
* Correspondence: cbzeng@git.edu.cn

**Featured Application: Typical applications include industrial automatic inspection, smart vehicles, text retrieval, and advanced human computer interfaces.**

**Abstract:** Scene text spotting has made tremendous progress with the in-depth research on deep convolutional neural networks (DCNN). Previous approaches mainly focus on the spotting of arbitrary-shaped scene text, on which it is difficult to achieve satisfactory results on dense scene text containing various instances of bending, occlusion, and lighting. To address this problem, we propose an approach called Rwin-FPN++, which incorporates the long-range dependency merit of the Rwin Transformer into the feature pyramid network (FPN) to effectively enhance the functionality and generalization of FPN. Specifically, we first propose the rotated windows-based Transformer (Rwin) to enhance the rotation-invariant performance of self-attention. Then, we attach the Rwin Transformer to each level on our feature pyramids to extract global self-attention contexts for each feature map produced by the FPN. Thirdly, we fuse these feature pyramids by upsampling to predict the score matrix and keypoints matrix of the text regions. Fourthly, a simple post-processing process is adopted to precisely merge the pixels in the score matrix and keypoints matrix and obtain the final segmentation results. Finally, we use the recurrent neural network to recognize each segmentation region and thus achieve the final spotting results. To evaluate the performance of our Rwin-FPN++ network, we construct a dense scene text dataset with various shapes and occlusion from the wiring of the terminal block of the substation panel cabinet. We train our Rwin-FPN++ network on public datasets and then evaluate the performance on our dense scene text dataset. Experiments demonstrate that our Rwin-FPN++ network can achieve an F-measure of 79% and outperform all other methods in F-measure by at least 2.8%. This is because our proposed method has better rotation invariance and long-range dependency merit.

**Keywords:** dense scene text spotting; segmentation; transformer; Rwin-FPN++





## 1. Introduction

Scene text spotting, i.e., detection and recognition of scene text in a unified network, is one of the most valuable tasks in the field of computer vision. Typical applications include industrial automatic inspection, smart vehicles, text retrieval, and advanced human–computer interfaces. In the past few years, convolutional neural network (CNN)-based scene text spotting has made remarkable progress. Jaderberg et al. [1] first adopted the CNN to detect and recognize scene text. Borisyuk et al. [2] employed Faster-RCNN [3] to detect scene text and a CTC loss [4] to recognize text. Liu et al. [5] proposed an adaptive bezier-curve network to detect and recognize curved scene text in real-time.

In recent years, segmentation-based spotting approaches have achieved great performance on arbitrary-shaped scene text. Zhou et al. [6] adopted the fully convolutional networks (FCN) [7] to segment the text regions and predict their contours. Wang et al. [8]

modified the feature pyramid networks (FPN) [9] to predict the segmentation results for each scene text region, and then used the PSE algorithm to compute the final detection result. In order to speed up the detection speed of the PSE algorithm, Wang et al. [10] proposed the text kernel representation method to spot the scene text. However, if these methods is applied to the dense scene text containing various instances of bending, occlusion, lighting, and other difficult situations, they struggle to achieve satisfactory results.

Recently, Transformer [11]-based network architecture has been proposed to establish self-attention mechanisms. Compared with CNN-based approaches, Transformer can parallel model global contexts of sequences and has achieved great progress in machine translation and natural language processing. More recently, Transformer has been adapted for computer vision and achieved state-of-the-art performances in image classification [12,13], medical image segmentation [14], and object detection [13]. However, if Transformer is directly used to encode the tokenized image patches for segmentation and detection tasks, the results are usually unsatisfactory [14]. This is due to the fact that Transformer cannot extract local low-level visual cues well, which can be compensated by CNN architectures (e.g., U-Net [15] and FPN [9]).

Inspired by the success of Swin Transformer [13] on object detection tasks, we propose a method called Rwin-FPN++, which incorporates the long-range dependency merit of Rwin Transformer into the feature pyramid networks (FPN) to effectively enhance the functionality and generalization of FPN. Specifically, we first propose the rotated windows-based Transformer (Rwin) to enhance the rotation-invariant performance of self-attention. Then, we attach the Rwin Transformer to each level on our feature pyramids to extract global self-attention contexts for each feature map produced by the FPN. Thirdly, we fuse these feature pyramids by upsampling to predict the score matrix and keypoints matrix of the text regions. Fourthly, a simple post-processing process is adopted to precisely merge the pixels in the score matrix and keypoints matrix and obtain the final segmentation results. Finally, we use the recurrent neural network to recognize each segmentation region, and thus achieve the final spotting results. To evaluate the performance of our Rwin-FPN++ network, we construct a dense scene text dataset with various shapes and occlusions from the substation secondary circuit cabinet wiring site. We train our Rwin-FPN++ network on public datasets and then evaluate the performance on our dense scene text dataset (Figure 1). The experiments demonstrate that our Rwin-FPN++ network can achieve better spotting performance for dense scene text compared with state-of-the-art approaches.

Our main contributions are as follows:

(1) We improve the Swin Transformer network [13] to the Rwin Transformer network. Compared with the shifted windows-based Transformer (Swin Transformer), the rotated windows-based Transformer (Rwin Transformer) can achieve better rotational invariance of the self-attention mechanism. For the task of scene text detection, because there are a large number of rotated and distorted texts, we modified the Swin Transformer by adding a rotating window self-attention mechanism. Thus our network can enhance the attention to rotated and distorted scene text.

(2) We combine the Rwin Transformer with the feature pyramid network to detect and recognize dense scene text. The Rwin Transformer is used to enhance the rotational invariance of the self-attention mechanism. The feature pyramid network is adopted to extract local low-level visual cues of scene text.

(3) A dense scene text dataset was constructed to evaluate the performance of our Rwin-FPN++ network. The 620 pictures of this dataset were taken from the wiring of the terminal block of the substation panel cabinet. Text instances in these pictures are very dense, with horizontal, multi-oriented, and curved shapes. This dataset can be downloaded from https://github.com/cbzeng110/-DenseTextDetection (accessed on 10 February 2022).

(4) The experiments show that our Rwin-FPN++ network can achieve an F-measure of 79% on our dense scene text. Compared with previous approaches, our method

outperforms all other methods in F-measure by at least 2.8% and achieves state-of-the-art spotting performances.

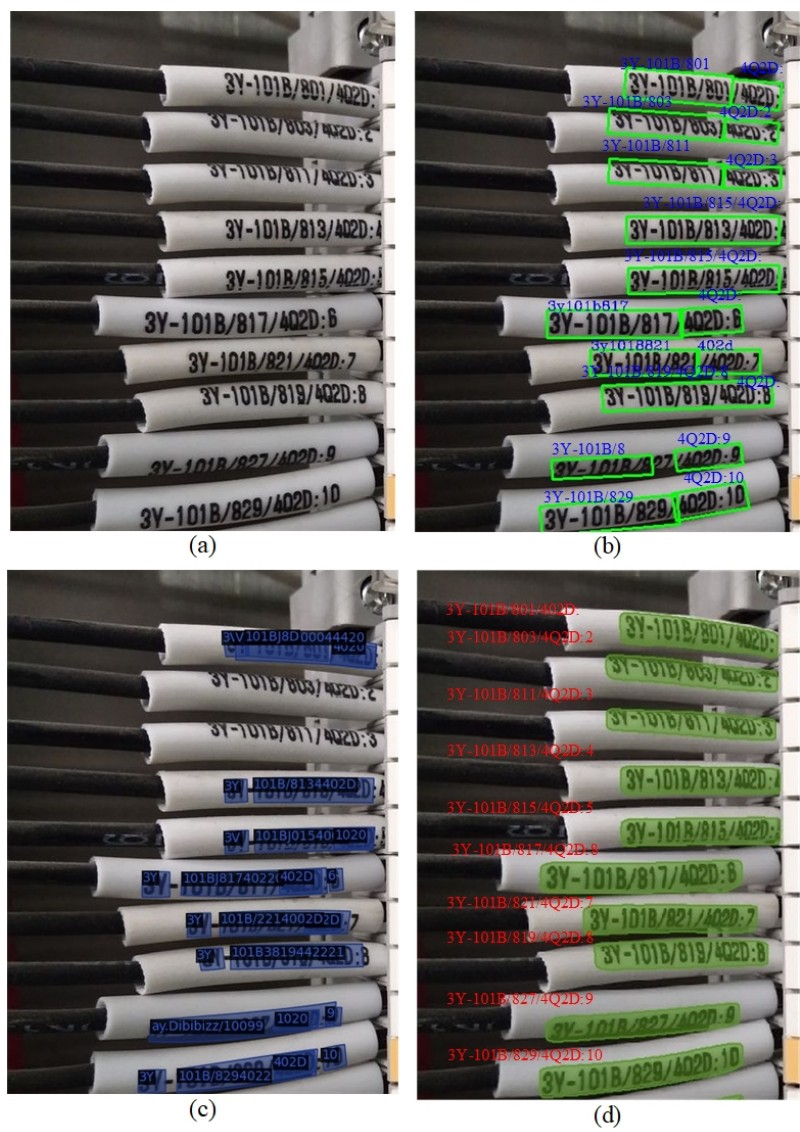

**Figure 1.** Our spotting results compared with state-of-the-art approaches for the dense scene text image. (**a**) is the input image. (**b**) is the result of Pan++ [10]. (**c**) is the result of ABCNet v2 [5]. (**d**) is the result of our Rwin-FPN++ network.

## 2. Related Work

We will describe the related work from three aspects, including scene text detection, scene text recognition, and scene text spotting.

**Scene Text Detection.** In the past few years, great progress has been achieved for deep learning-based scene text detection. Zhou et al. [6] adopted fully convolutional networks (FCN) [7] to segment the text regions and predict their contours. Wang et al. [8] modified feature pyramid networks (FPN) [9] to predict the segmentation results for each scene text region, and then used the PSE algorithm to compute the final detection result. Liao et al. [16] modified the SSD [17] algorithm to detect arbitrary-shaped scene text. Zhu et al. [18] detected text regions in the Fourier domain and proposed a Fourier Contour Embedding (FCE) method to represent curved text contours. Dai et al. [19] proposed a progressive contour regression approach to detect various aspect ratios of scene texts. Recently, Transformer has been adapted for computer vision and achieved state-of-the-art performances in image classification [12,13] and object detection [13]. Tang et al. [20] proposed

a simple and effective Transformer-based scene text detection network, which is mainly composed of a feature sampling module and a feature combination module. However, if Transformer is directly used to encode the tokenized image patches for segmentation and detection tasks, the results are usually unsatisfactory [14]. This is due to the fact that Transformer cannot extract local low-level visual cues well, which can be compensated by the CNN architectures (e.g., U-Net [15] and FPN [9]).

**Scene Text Recognition.** The task of scene text recognition is to identify the text content of the segmented scene text area. Shi et al. [21] adopted RNN to extract visual feature sequences produced by the CNN and achieved highly competitive performances on scene text recognition. Qiao et al. [22] recognized the low-quality scene texts robustly using an enhanced encoder-decoder network. Aberdam et al. [23] recognized the scene text by extending the contrastive learning methods. Fang et al. [24] proposed a bidirectional and autonomous ABINet to recognize the scene text.

**Scene Text Spotting.** The purpose of scene text spotting is to detect and recognize scene text in a unified network. Jaderberg et al. [1] first adopted the CNN to detect and recognize the scene text. Liao et al. [16] modified the SSD [17] algorithm to detect arbitrary-shaped scene text. Borisyuk et al. [2] employed Faster-RCNN [3] to detect scene text and a CTC loss [4] to recognize text. Borisyuk et al. [25] proposed a fully convolutional point-gathering network (PGNet) to recognize multi-oriented text instances in real-time. Wang et al. [10] treated the text line as the text kernel and proposed an end-to-end network for curved text spotting. Liu et al. [5] proposed an adaptive bezier-curve network to detect and recognize arbitrary-shaped scene text in real-time. However, these methods mainly focus on the spotting of arbitrary-shaped scene text, which is difficult to achieve satisfactory results on dense scene text containing various instances of bending, occlusion, and lighting.

## 3. Proposed Method

### 3.1. Overall Architecture

A high-level overview of the proposed architecture is illustrated in Figure 2. First, we detect the text regions using the proposed Rwin Transformer-based feature pyramid network. Then, we segment the detected multiple text regions from the background. Finally, we employ the convolutional recurrent neural network to recognize each segmented region and thus achieve the final spotting results.

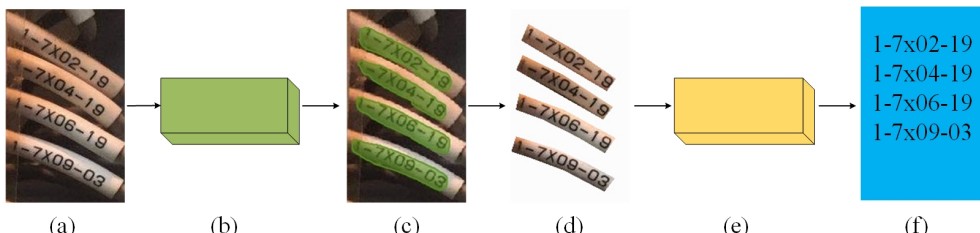

**Figure 2.** The overall architecture of Rwin-FPN++. (**a**) is the input image with dense scene text. (**b**) is the Rwin Transformer-based feature pyramid network. (**c**) is the result of text detection. (**d**) is the result of text segmentation. (**e**) is the convolutional recurrent neural network to recognize each segmentation region. (**f**) is the final spotting result.

### 3.2. Text Detection

Figure 3 shows the detection branch of Rwin-FPN++. We adopt the segmentation=based strategy to detect dense scene text. The backbone of our network is ResNet [26]. We attach the Rwin Transformer to each layer of ResNet to extract global self-attention contexts of feature maps. Then, the low-level texture features and the high-level semantic feature maps are connected to the Feature Fusion and Project Module (FFPM). Then, we project the FFPM into the score matrix and keypoints matrix of the text regions. Finally, we merge the values in the score matrix and keypoints matrix and thus obtain the final segmentation result.

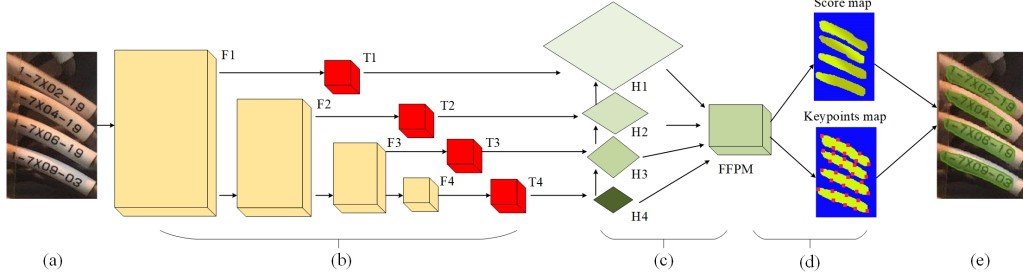

**Figure 3.** The overall pipeline of detection. (**a**) is the input image with dense scene text. (**b**) At each layer of the feature pyramid network, the Rwin Transformer is adopted to model the rotational invariance of the self-attention mechanism. (**c**) The four layers of the feature pyramid network (FPN) are combined into a new Module (FFPM). (**d**) The module FFPM produces the text score matrix and keypoints matrix. (**e**) We merge the values in the score matrix and keypoints matrix and obtain the final segmentation result.

### 3.2.1. Rwin Transformer

In recent years, Transformer has been adapted for computer vision and achieved state-of-the-art performances in image classification [12,13] and object detection [13]. Inspired by the success of Swin Transformer [13] on object detection tasks, we propose a method called Rwin-FPN++, which incorporates the long-range dependency merit of Rwin Transformer into the feature pyramid networks (FPN) to effectively enhance the functionality and generalization of FPN. Swin Transformer proposes a self-attention mechanism based on the shifted windows. First, the self attention of the window is calculated, and then the self attention after the shifted window is carried out. In this way, communication between windows is realized, thus achieving the effect of global modeling and better translation invariance. For the task of scene text detection, because there are a large number of rotated and distorted texts, we modified the Swin Transformer by adding a rotating window self-attention mechanism. Thus our Rwin Transformer can enhance the attention to rotated and distorted scene text.

The main idea of the Rwin Transformer is to compute the multi-head self-attention (MSA) on regular and rotated window partitions consecutively. Following the design of [13], the Rwin Transformer block is composed of the windows-based MSA (W-MSA) module and the rotated windows-based MSA (RW-MSA) module. Both W-MSA and RW-MSA are composed of four layers of windows-based MSA and two layers of Multi-Layer Perceptron (MLP) with GELU operation. Layernorm (LN) is calculated between each MSA module and each residual connection. The Rwin Transformer block can be computed as follows:

$$\hat{z_1} = W\_MSA(LN(z_0)) + z_0 \tag{1}$$

$$z_1 = MLP(LN(\hat{z_1})) + \hat{z_1} \tag{2}$$

$$\hat{z_2} = RW\_MSA(LN(z_1)) + z_1 \tag{3}$$

$$z_2 = MLP(LN(\hat{z_2})) + \hat{z_2} \tag{4}$$

where $z_0$, $\hat{z_1}$, $z_1$, $\hat{z_2}$ and $z_2$ represent the output of the encoder, W_MSA, MLP, RW_MSA, and MLP, respectively.

The MSA block is computed as follows:

$$MultiHead(Q, K, V) = Concat(head_1, \ldots, head_n) \tag{5}$$

$$head_i = SoftMax\left(\frac{QK^T}{\sqrt{d}} + B\right)V \tag{6}$$

where $K$, $Q$, and $V$ represent the key, query, and value matrices; $d$ is the dimension of query and key, and $B$ is the relative position bias; Concat denotes the concatenation operation.

Following [13], we partition each output layer produced by Resnet into multiple windows of size $4 \times 4$. The windows are then reshaped into a sequence of 1D patches, which are treated as the token of the Transformer. These embedded patches are then mapped into three matrices ($Q$, $K$, $V$ in Equation (6)) through a linear operation. Then one attention module can be computed using Equation (6), and the multi-head self-attention (MSA) block can be obtained using Equation (5). The residual output $\hat{z}_2$ can thus be calculated using Equation (3). Finally, the global self-attention output $z_2$ is obtained using Equation (4).

### 3.2.2. Network Design

The backbone of our Rwin-FPN++ network is the feature extraction part of the 50-layer ResNet. We first use the ResNet to extract four levels of feature maps (denoted as $F_1$, $F_2$, $F_3$, and $F_4$, see Figure 3b) for the input image. Then, we attach the Rwin Transformer block (denoted as $T_1$, $T_2$, $T_3$, $T_4$) to each layer of ResNet to extract the global self-attention contexts of feature maps. Finally, we use the upsampling strategy described in EAST [6] to gradually merge these four feature maps.

$$G_i = \begin{cases} upsampling(H_i) & if \quad i < 3 \\ \\ smooth(H_i) & if \quad i = 4 \end{cases} \tag{7}$$

$$H_i = \begin{cases} F_i + T_i & if \quad i = 1 \\ \\ smooth(F_i + T_i) & otherwise \end{cases} \tag{8}$$

where $G_i$ denotes the merge operation, $H_i$ is the concatenation of the feature map and Rwin Transformer block, and the smooth operator refers to the (Conv(3,3), Layer Normalization [27], Relu) operation.

Then, we transform the merged feature maps with the smooth operation and produce the score matrix and keypoints matrix of the text regions. Finally, we use polygons to connect keypoints located in each connected region in the score matrix and set the pixels in each polygon area as the foreground. For each of the pixels outside the polygon in the score matrix, if the score of the pixel is greater than the threshold, it is set as the foreground; otherwise, it is set as the background.

### 3.2.3. Detection Loss Function

The loss function for training our Rwin-FPN++ network can be formulated as follows:

$$L = \lambda L_s + (1 - \lambda) L_k \tag{9}$$

where $L_s$ and $L_k$ denote the losses for the score matrix and the keypoints matrix, respectively. $\lambda$ is the balance factor between two losses, and is set as $\lambda = 0.7$. We use average binary cross-entropy loss to compute the $L_s$, and mean squared error (MSE) to compute the $L_k$.

### 3.3. Text Recognition

The recognition branch of Rwin-FPN++ is illustrated in Figure 4. We adopt a network structure similar to CRNN [21] to recognize scene text. For each segmented text region, we first extract the convolutional feature sequence using the 18-layer ResNet. Then, we adopt the GRU [28] as the recurrent network to predict each frame of the convolutional feature sequence. Thirdly, the lexicon-free transcription layer is built to translate the per-frame predictions by the GRU into a label sequence. The transcription layer is composed of the conditional probability defined in the Connectionist Temporal Classification (CTC) layer [4]. The whole recognition network can be jointly trained with the CTC loss function [4].

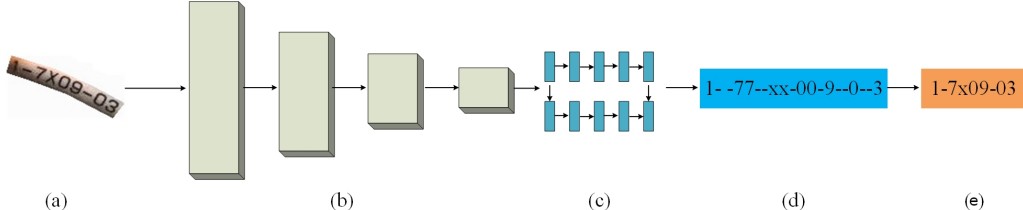

**Figure 4.** The overall pipeline of recognition. (**a**) is the input image. (**b**) is the convolutional layers. (**c**) is the recurrent layers. (**d**) is the lexicon-free transcription layer. (**e**) is the final recognition result.

## 4. Experiment

### 4.1. Training Set

**SynthText** [29] contains approximately 8 million synthetic multi-oriented text instances. Each text instance is labeled at word-level bounding boxes and transcriptions at the word level.

**CTW1500** [30] consists of 1500 images that are harvested from the internet and some image libraries. This dataset provides 10,751 cropped text instances, including 3530 curved and multi-oriented text instances.

**Total-Text** [31] contains 1555 images with multiple curved, multi-oriented, and horizontal text regions. All text instances are labeled with eight vertices of the quadrangle, as well as corresponding transcriptions.

**ICDAR 2017 MLT** [32] is commonly used for arbitrarily shaped (i.e., horizontal, multi-oriented, and curved) text spotting. This dataset consists of 7200 images for training, 1800 images for validation, and 9000 images for evaluation. This dataset contains a lot of scene text with arbitrary orientations. All of the text instances are labeled with eight vertices of the quadrangle and transcriptions at the word level.

**ICDAR 2019 ART** [33] includes 5603 images for training and 4563 images for testing. This dataset is aimed at introducing the multi-oriented scene text problem to the research field of complicated scene text. All of the text instances are annotated with 10 vertices of the quadrangle and transcriptions at the word level.

### 4.2. Implementation Details

Our Rwin-FPN++ network is implemented in Pytorch on two NVIDIA Quadro RTX8000 GPUs. We use the Resnet trained with the STKM method [34] to initialize the feature extractor stem of Rwin-FPN++. This step can help the training process converge faster and generate higher-quality segmentation results. Then, the Rwin-FPN++ network is end-to-end trained using the ADAM optimizer [35], and the initial learning rate is set to 0.01, which decays to one-tenth every 3k iterations. The input images are resized to $380 \times 960$ for both training and evaluation.

Following [10], we choose the SynthText [29] dataset to pre-train our Rwin-FPN++ network. Then, we refine our model on five public datasets, including CTW1500 [30], Total-Text [31], ICDAR 2015 [36], ICDAR 2017 MLT [32], and ICDAR 2019 ART [33]. The total number of training images in these five datasets is 9658. The text regions were annotated by 4, 8, and 12 vertices of the polygons. To generate the ground truth of the score matrix, we set the score value of pixels within these polygonal areas to 1. Otherwise, they are set to 0.

### 4.3. Evaluation Results

To evaluate the performance of our Rwin-FPN++ network, a dense scene text dataset with 620 pictures was taken from the wiring of the terminal block of the substation panel cabinet. Text instances in these pictures are very dense, with horizontal, multi-oriented, and curved shapes. (Figure 5). All these text instances were annotated with an eight-vertices polygon bounding box, as well as the corresponding transcriptions.

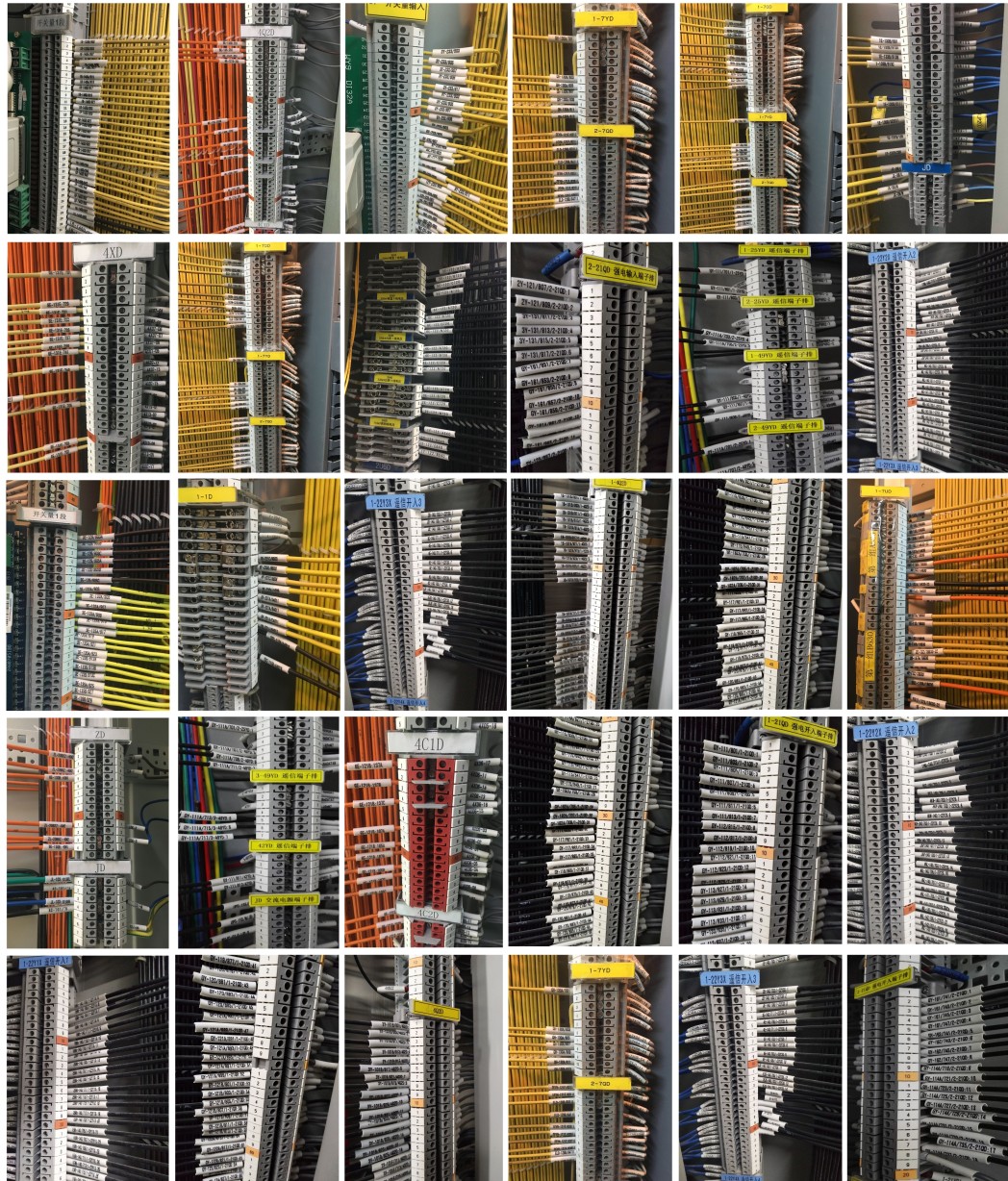

**Figure 5.** Some examples of the dense scene text dataset from the wiring of the terminal block of the substation panel cabinet.

### 4.3.1. Evaluation of Detection Task

We use the standard recall, precision, and F-measure metrics to evaluate the performance of detection. We have compared six state-of-the-art methods with the proposed RWIN-FPN++ approach. All six methods have publicly released code available for download. We retrain these six approaches with the same protocol described in Section 4.1. Table 1 shows that our Rwin-FPN++ method can achieve an F-measure of 79% and outperform all other methods in F-measure by at least 2.8%. These results prove that our Rwin-FPN++ approach can detect the dense scene text accurately. It is worth noting that the F-measure of all methods are lower than that of the five public datasets above, which indicates that our dense scene text is more difficult than the current public datasets. Figure 6 shows some qualitative comparisons of detection performance between state-of-the-art methods and the Rwin-FPN++. Compared with state-of-the-art methods, our Rwin-FPN++ network is more effective at detecting dense scene texts with various bends, occlusion, and lighting. Figure 7 shows other detection results of the Rwin-FPN++ network.

**Table 1.** Detection comparison with state-of-the-art methods on our dense scene text dataset.

| Method | Precision | Recall | F-Measure |
|---|---|---|---|
| EAST [6] | 53.2 | 48.5 | 50.7 |
| Textspotter [37] | 72.4 | 70.9 | 71.6 |
| Textboxes++ [38] | 73.2 | 71.9 | 72.6 |
| PSENet [8] | 73.7 | 70.1 | 71.8 |
| Pan++ [10] | 76.6 | 71.1 | 73.7 |
| ABCNet v2 [5] | 79.7 | 73.1 | 76.2 |
| Our approach | **82.2** | **76.1** | **79.0** |

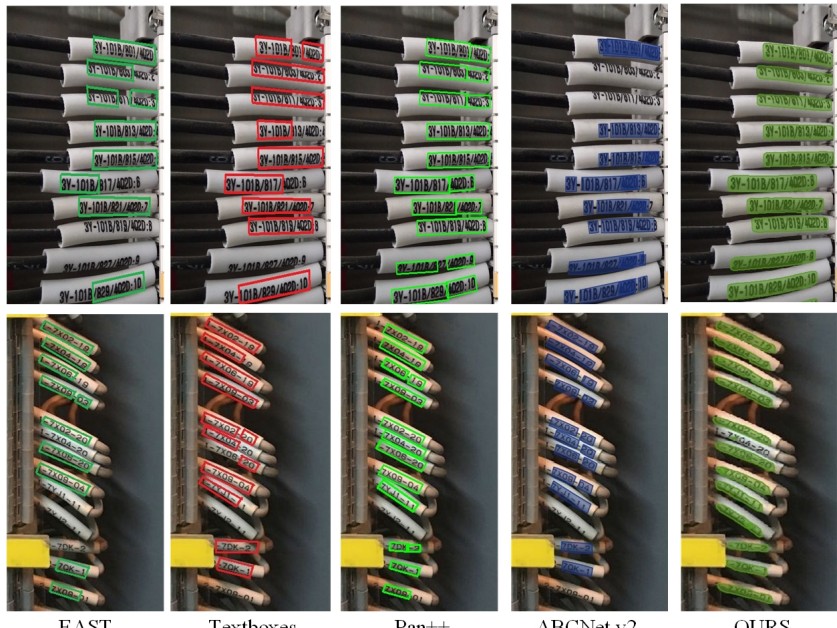

EAST　　　　Textboxes　　　　Pan++　　　　ABCNet v2　　　　OURS

**Figure 6.** Qualitative comparisons of detection performance between state-of-the-art methods and the Rwin-FPN++. Compared with state-of-the-art methods, our Rwin-FPN++ network is more effective at detecting dense scene texts with various bends, occlusion, and lighting.

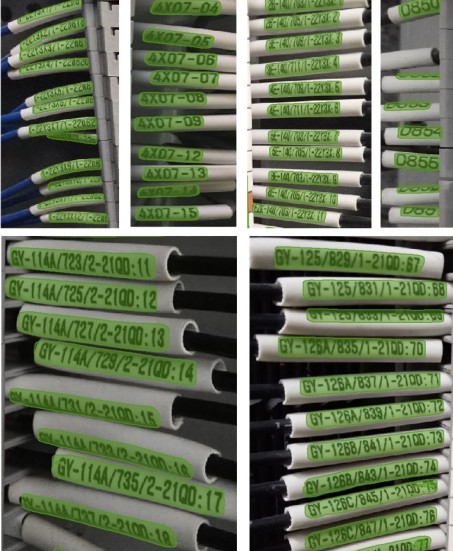

**Figure 7.** Some other detection results of the Rwin-FPN++ network.

We also compare our method to previous methods on the public dataset Total-Text [31] and CTW1500 [30]. Total-Text contains 1555 images with multiple curved, multi-oriented,

and horizontal text regions; 1255 images were used for training and the other 300 images for testing. All the text instances are labeled with eight vertices of the quadrangle, as well as corresponding transcriptions. CTW1500 [30] consists of 1500 images, which are harvested from the internet and some image libraries. This dataset provides 10,751 cropped text instances, including 3530 curved and multi-oriented text instances. Tables 2 and 3 show that our Rwin-FPN++ method can achieve an F-measure of 88% and 85.7% and outperform all state-of-the-art methods. This result shows that our method is not only suitable for the detection of dense scene text but also can achieve good results for the detection of curved and multi-oriented scene text.

**Table 2.** Detection comparison with state-of-the-art methods on the Total-Text dataset.

| Method | Precision | Recall | F-Measure |
|---|---|---|---|
| East [6] | 50.0 | 36.2 | 42.0 |
| Textboxes++ [38] | 77.2 | 81.9 | 79.6 |
| PSENet [8] | 78.0 | 84.0 | 80.9 |
| Pan++ [10] | 81.0 | 89.3 | 85.0 |
| ABCNet v2 [5] | 84.1 | 90.2 | 87.0 |
| Our approach | **85.2** | **91.2** | **88.0** |

**Table 3.** Detection comparison with state-of-the-art methods on the CTW1500 dataset.

| Method | Precision | Recall | F-Measure |
|---|---|---|---|
| East [6] | 48.0 | 35.2 | 40.6 |
| Textboxes++ [38] | 74.1 | 80.5 | 77.2 |
| PSENet [8] | 76.0 | 82.2 | 78.9 |
| Pan++ [10] | 80.0 | 83.5 | 81.7 |
| ABCNet v2 [5] | 83.8 | 85.6 | 84.5 |
| Our approach | **83.1** | **88.5** | **85.7** |

### 4.3.2. Evaluation of Spotting Task

We then evaluate the end-to-end text spotting performance of our approach with state-of-the-art methods on our dense scene text dataset. We compare the spotting results in four modes: lexicon-free, strong lexicon, weak lexicon, and generic lexicon. Table 4 shows that our Rwin-FPN++ method can achieve a spotting score of 73.2% and outperform all other methods in spotting score by at least 3.1%. These results prove that our Rwin-FPN++ approach can recognize the dense scene text accurately. Thus, our proposed approach can be well applied to the intelligent inspection of substations. Figure 8 shows the spotting results of the Rwin-FPN++ network.

**Table 4.** End-to-end text spotting comparison with state-of-the-art methods on our dense scene text dataset. "None" represents lexicon-free. "S", "W", and "G" represent recognition with Strong, Weak, and Generic lexicon, respectively.

| Method | None | S | W | G |
|---|---|---|---|---|
| Mask TextSpotter [39] | 70.5 | 82.2 | 81.0 | 69.0 |
| Textboxes++ [38] | 67.2 | 81.9 | 79.6 | 67.2 |
| Craft [40] | 68.0 | 82.1 | 80.9 | 68.0 |
| Pan++ [10] | 71.0 | 83.3 | 82.0 | 69.1 |
| ABCNet v2 [5] | 72.1 | 85.2 | 83.0 | 70.1 |
| Our approach | **75.2** | **88.5** | **85.1** | **73.2** |

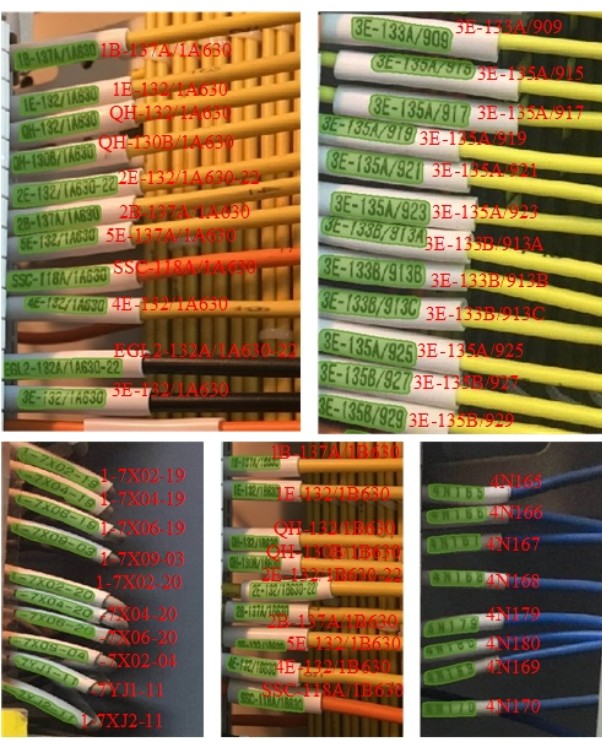

**Figure 8.** Some spotting results of Rwin-FPN++.

## 5. Ablation Study

We present ablation experiments to analyze how the Rwin Transformer contributes to the final performance of our dense scene text dataset. We first train the detection network without Rwin Transformer on the dense scene text dataset. Then, the detection network with Rwin Transformer is trained on the same set. We also train the detection network with Swin Transformer on the same set and compare the performance of the three frameworks. Table 5 shows a substantial improvement for all the metrics by integrating the Rwin Transformer into the detection network. The improvement of these metrics demonstrates that the Rwin Transformer can extract global self-attention contexts and thus achieve better visual quality. Furthermore, compared with the shifted windows-based Transformer (Swin Transformer [13]), the rotated windows-based Transformer (Rwin) can outperform Swin Transformer in F-measure by at least 2.1%. This result demonstrates that Rwin Transformer can achieve better rotation-invariant performance.

**Table 5.** Performance comparison of network architectures.

| Method | Precision | Recall | F-Measure |
| --- | --- | --- | --- |
| without Transformer | 75.0 | 66.2 | 70.3 |
| Swin Transformer | 80.2 | 73.9 | 76.9 |
| Rwin Transformer | **82.2** | **76.1** | **79.0** |

## 6. Conclusions

In this paper, we propose the Rwin-FPN++ network to implement the task of dense scene text spotting. We improve the Swin Transformer to the Rwin Transformer to achieve better rotation-invariant performance and long-range dependency merit. Then, we incorporate the Rwin Transformer into the FPN to detect and recognize dense scene text simultaneously. The FPN is designed to extract local low-level visual cues, and the Rwin Transformer is adopted to model the global contexts of the scene text. Finally, we evaluate the performance of our dense scene text dataset, which is collected from the wiring of the terminal block of the substation panel cabinet. Experiments demonstrate that our

Rwin-FPN++ network can achieve an F-measure of 79% and outperform all other methods in F-measure by at least 2.8% on our dense scene text dataset. Furthermore, our Rwin-FPN++ method can achieve an F-measure of 88% and 85.7% on the public datasets Total-Text [31] and CTW1500 [30] and outperform all state-of-the-art methods. These results demonstrate that our Rwin-FPN++ network has better rotation invariance and long-range dependency merit.

### 7. Future Work

Although our Rwin-FPN++ network has achieved a state-of-the-art spotting effect, it still has the following limitations: (1). Four Rwin Transformers are used in our network structure, which makes the network have more parameters and increases the network calculation amount. Thus the final recognition speed of our system is only 8 FPS, which does not meet the requirements of real-time computing. In the future, we will continue to optimize the network structure to achieve real-time computing capabilities. (2). The detection branch and recognition branch of our network are calculated separately, and there is no shared feature extraction between each other, resulting in some unnecessary repeated calculations. In the future, we will improve our network structure so that detection and recognition can share features. Thus the spotting speed can be improved further. (3). We intend to port the Rwin-FPN++ approach into the mobile devices and serve the automatic inspection of substations.

**Author Contributions:** C.Z. conceived the Rwin-FPN++ algorithm and Y.L. and C.Z. conducted the experiment(s). C.S. analyzed the results and wrote the original draft. All authors reviewed the manuscript and agreed to the published version of the manuscript.

**Funding:** This research was funded by the National Natural Science Foundation of China (Grant No. 61966006), the Guizhou Provincial Science and Technology Projects (Grant No. [2020]1Y281), and the Science Research Foundation for High-level Talents of Guizhou Institute of Technology (Grant No. XJGC20150108).

**Institutional Review Board Statement:** Ethical review and approval were waived for this study due to our research does not involve humans or animals.

**Data Availability Statement:** A dense scene text dataset was constructed to evaluate the performance of our Rwin-FPN++ network. The 620 pictures in this dataset were taken from the wiring of the terminal block of the substation panel cabinet. Text instances in these pictures are very dense, with horizontal, multi-oriented, and curved shapes. This dataset can be downloaded from https://github.com/cbzeng110/-DenseTextDetection (accessed on 10 February 2022).

**Conflicts of Interest:** The authors declare no conflict of interest. The funders had no role in the design of the study.

### Abbreviations

The following abbreviations are used in this manuscript:

| | |
|---|---|
| DCNN | Deep Convolutional Neural Network(CNN) |
| Rwin | Rotated Windows-based Transformer |
| Swin | Shifted Windows-based Transformer |
| FPN | Feature Pyramid Network |
| Rwin-FPN++ | Rwin Transformer with FPN for Dense Scene Text Spotting |

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
