# Peer review of "Rwin-FPN++: Rwin Transformer with Feature Pyramid Network for Dense Scene Text Spotting"

_applsci, doi:10.3390/app12178488_

Round 1

Reviewer 1 Report

·       Please describe results in the abstract section. No results are shown. Also try to explain how your results are superior to others work.

·       List of contributions are too long and difficult to understand. Make each contribution in one sentence.

·       Introduction section should end with the description of paper design and flow of sections.

·       Relates work should focus on recent studies from 2021 and 2022.

·       Methodology should be properly be supported by the parameter description.

·       Dataset section should come before methodology section. It should not be in results section.

·       All the preprocessing, training and testing details should be described separately.

·       Results section should have a comparative analysis subsection wherein the authors should show how superior is their work compared to other works in tabular format.

·       Conclusion section is very week. Please try to work on this section and enhance it further.

·       Future direction can be a separate section. Also, limitations of the current study should be discussed.

·       References are not in order. Please use reference management tool to cite the references.

Reviewer 2 Report

The authors propose an improved variant of the popular Swin Transformer for scene text spotting. Combined with a feature pyramid network and an RNN, authors present experimental results on one public dataset and one private dataset to show that the proposed method provides reasonable improvement. 

I am a little doubtful about the significance of the Rwin approach. Can't that be achieved simply by augmenting the original data with a rotational variant? It seems redundant to include a rotational variant in the transformer itself. 

I am not an expert in text processing, but the combination of transformer and RNN seems again, redundant. Shouldn't the authors use transformers end-to-end? 

The results presented show good improvement, but I am not sure why the authors presented test results on only one public dataset, when they trained the model on multiple of them. 

I hope the authors can address these comments through a revision. 

Reviewer 3 Report

In this paper the authors are proposing a method for scene text spotting combining several techniques. They propose a method they called Rwin-FPN++ that incorporates the rotated windows transformers (Rwin) with feature pyramid network (FPN). They use additional processing and post processing technique to improve the results. The research is in the scope of the journal and could be interesting to its readers. However, the manuscript needs additional work, especially with respect to clarification on the novelty and main contributions. Details on the implementation are needed. Please consider if the research can be repeated based on the paper.

Comments/Questions:

  • Introduction would benefit from additional paragraph on motivation and problems that can be solved with this method.

  • While the proposed method section provides lots of details, it is not clear which parts of the method are novel, i.e. the novelty and main contribution of this research needs to be clearly stated and explained.

  • Also, what was the process of deriving the method, i.e. how did you decide to select and combine these techniques. What was the criteria to select these techniques that were combined?

  • As for the datasets, used, the authors do provide info on the data and its sources, but it is not clear how the data was used and preprocessed for the use. If the readers (other researchers) would like to repeat this research, I am not sure enough information is provided and the same results could be achieved; more details on the dataset preparation and implementation would be needed. You may need to add additional table addressing the dataset preparation and/or a flow chart explaining the implementation.

  • In the conclusions, please consider using bullets to clearly itemize the main contributions and results of the study

Round 2

Reviewer 1 Report

The authors have responded to my comments..

Author Response

Thank the reviewer for your careful and valuable comments. The authors have responded to the reviewer's comments in round 1.

Reviewer 2 Report

Authors have reasonably addressed my comments. 

Author Response

Thank the reviewer for your careful and valuable comments. We have responded to the reviewer's comments in round 1.

Reviewer 3 Report

I would suggest change to Conclusions to add the summary of the main results and performance of the proposed algorithm.
